# Scalable Representation Learning in Linear Contextual Bandits with Constant Regret Guarantees

**Andrea Tirinzoni**
Meta
tirinzoni@meta.com

**Matteo Papini**
Universitat Pompeu Fabra
matteo.papini@upf.edu

**Ahmed Touati**
Meta
atouati@meta.com

**Alessandro Lazaric**
Meta
lazaric@meta.com

**Matteo Pirotta**
Meta
pirotta@meta.com

## Abstract

We study the problem of representation learning in stochastic contextual linear bandits. While the primary concern in this domain is usually to find *realizable* representations (i.e., those that allow predicting the reward function at any context-action pair exactly), it has been recently shown that representations with certain spectral properties (called *HLS*) may be more effective for the exploration-exploitation task, enabling *LinUCB* to achieve constant (i.e., horizon-independent) regret. In this paper, we propose BANDITSRL, a representation learning algorithm that combines a novel constrained optimization problem to learn a realizable representation with good spectral properties with a generalized likelihood ratio test to exploit the recovered representation and avoid excessive exploration. We prove that BANDITSRL can be paired with any no-regret algorithm and achieve constant regret whenever an *HLS* representation is available. Furthermore, BANDITSRL can be easily combined with deep neural networks and we show how regularizing towards *HLS* representations is beneficial in standard benchmarks.

## 1   Introduction

The contextual bandit is a general framework to formalize the exploration-exploitation dilemma arising in sequential decision-making problems such as recommendation systems, online advertising, and clinical trials [e.g., 1]. When solving real-world problems, where contexts and actions are complex and high-dimensional (e.g., users' social graph, items' visual description), it is crucial to provide the bandit algorithm with a suitable representation of the context-action space. While several representation learning algorithms have been proposed in supervised learning and obtained impressing empirical results [e.g., 2, 3], how to *efficiently* learn representations that are effective for the exploration-exploitation problem is still relatively an open question.

The primary objective in representation learning is to find features that map the context-action space into a lower-dimensional embedding that allows fitting the reward function accurately, i.e., *realizable* representations [e.g., 4–10]. Within the space of realizable representations, bandit algorithms leveraging features of smaller dimension are expected to learn faster and thus have smaller regret. Nonetheless, Papini et al. [11] have recently shown that, even among realizable features, certain representations are naturally better suited to solve the exploration-exploitation problem. In particular, they proved that LINUCB [12, 13] can achieve constant regret when provided with a "good" representation. Interestingly, this property is not related to "global" characteristics of the feature map (e.g., dimension, norms), but rather on a spectral property of the representation (the space associated to optimal actions should cover the context-action space, see HLS property in Def. 2.1). This naturally

36th Conference on Neural Information Processing Systems (NeurIPS 2022).

raises the question whether it is possible to learn such representation at the same time as solving the contextual bandit problem. Papini et al. [11] provided a first positive answer with the LEADER algorithm, which is proved to perform as well as the best realizable representation in a given set up to a logarithmic factor in the number of representations. While this allows constant regret when a realizable HLS representation is available, the algorithm suffers from two main limitations: **1)** it is entangled with LINUCB and it can hardly be generalized to other bandit algorithms; **2)** it learns a different representation for each context-action pair, thus making it hard to extend beyond finite representations to arbitrary functional space (e.g., deep neural networks).

In this paper, we address those limitations through BANDITSRL, a novel algorithm that decouples representation learning and exploration-exploitation so as to work with any no-regret contextual bandit algorithm and to be easily extended to general representation spaces. BANDITSRL combines two components: 1) a representation learning mechanism based on a constrained optimization problem that promotes "good" representations while preserving realizability; and 2) a generalized likelihood ratio test (GLRT) to avoid over exploration and fully exploit the properties of "good" representations. The main contributions of the paper can be summarized as follows:

1. We show that adding a GLRT on the top of any no-regret algorithm enables it to exploit the properties of a HLS representation and achieve constant regret. This generalizes the constant regret result for LINUCB in [11] to any no-regret algorithm.

2. Similarly, we show that BANDITSRL can be paired with any no-regret algorithm and perform effective representation selection, including achieving constant regret whenever a HLS representation is available in a given set. This generalizes the result of LEADER beyond LINUCB. In doing this we also improve the analysis of the misspecified case and prove a tighter bound on the time to converge to realizable representations. Furthermore, numerical simulations in synthetic problems confirm that BANDITSRL is empirically competitive with LEADER.

3. Finally, in contrast to LEADER, BANDITSRL can be easily scaled to complex problems where representations are encoded through deep neural networks. In particular, we show that the Lagrangian relaxation of the constrained optimization problem for representation learning becomes a regression problem with an auxiliary representation loss promoting HLS-like representations. We test different variants of the resulting NN-BANDITSRL algorithm showing how the auxiliary representation loss improves performance in a number of dataset-based benchmarks.

## 2 Preliminaries

We consider a stochastic contextual bandit problem with context space $\mathcal{X}$ and finite action set $\mathcal{A}$. At each round $t \geq 1$, the learner observes a context $x_t$ sampled i.i.d. from a distribution $\rho$ over $\mathcal{X}$, selects an action $a_t \in \mathcal{A}$, and receives a reward $y_t = \mu(x_t, a_t) + \eta_t$ where $\eta_t$ is a zero-mean noise and $\mu : \mathcal{X} \times \mathcal{A} \to \mathbb{R}$ is the expected reward. The objective of a learner $\mathfrak{A}$ is to minimize its pseudo-regret $R_T := \sum_{t=1}^{T} \left( \mu^\star(x_t) - \mu(x_t, a_t) \right)$ for any $T \geq 1$, where $\mu^\star(x_t) := \max_{a \in \mathcal{A}} \mu(x_t, a)$. We assume that for any $x \in \mathcal{X}$ the optimal action $a_x^\star := \operatorname{argmax}_{a \in \mathcal{A}} \mu(x, a)$ is unique and we define the gap $\Delta(x, a) := \mu^\star(x) - \mu(x, a)$. We say that $\mathfrak{A}$ is a no-regret algorithm if, for any instance of $\mu$, it achieves sublinear regret, i.e., $R_T = o(T)$.

We consider the problem of representation learning in given a candidate function space $\Phi \subseteq \left\{ \phi : \mathcal{X} \times \mathcal{A} \to \mathbb{R}^{d_\phi} \right\}$, where the dimensionality $d_\phi$ may depend on the feature $\phi$. Let $\theta_\phi^\star = \operatorname{argmin}_{\theta \in \mathbb{R}^{d_\phi}} \mathbb{E}_{x \sim \rho} \left[ \sum_a (\phi(x, a)^\mathsf{T} \theta - \mu(x, a))^2 \right]$ be the best linear fit of $\mu$ for representation $\phi$. We assume that $\Phi$ contains a linearly realizable representation.

**Assumption 1** (Realizability). *There exists an (unknown) subset $\Phi^\star \subseteq \Phi$ such that, for each $\phi \in \Phi^\star$, $\mu(x, a) = \phi(x, a)^\mathsf{T} \theta_\phi^\star, \forall x \in \mathcal{X}, a \in \mathcal{A}$.*

**Assumption 2** (Regularity). *Let $\mathcal{B}_\phi := \{ \theta \in \mathbb{R}^{d_\phi} : \|\theta\|_2 \leq B_\phi \}$ be a ball in $\mathbb{R}^{d_\phi}$. We assume that, for each $\phi \in \Phi$, $\sup_{x,a} \|\phi(x, a)\|_2 \leq L_\phi$, $\|\theta_\phi^\star\|_2 \leq B_\phi$, $\sup_{x,a} |\phi(x, a)^\mathsf{T} \theta| \leq 1$ for any $\theta \in \mathcal{B}_\phi$ and $|y_t| \leq 1$ almost surely for all $t$. We assume parameters $L_\phi$ and $B_\phi$ are known. We also assume the minimum gap $\Delta = \inf_{x \in \mathcal{X}: \rho(x) > 0, a \in \mathcal{A}, \Delta(x,a) > 0} \{\Delta(x, a)\} > 0$ and that $\lambda_{\min} \left( \frac{1}{|\mathcal{A}|} \sum_a \mathbb{E}_{x \sim \rho}[\phi(x, a)\phi(x, a)^\mathsf{T}] \right) > 0$ for any $\phi \in \Phi^\star$, i.e, all realizable representations are non-redundant.*

Under Asm. 1, when $|\Phi| = 1$, the problem reduces to a stochastic linear contextual bandit and can be solved using standard algorithms, such as LINUCB/OFUL [12, 13], LinTS [14], and $\epsilon$-greedy [15], which enjoy sublinear regret and, in some cases, logarithmic problem-dependent regret. Recently, Papini et al. [11] showed that LINUCB only suffers constant regret when a *realizable* representation is HLS, i.e., when the features of optimal actions span the entire $d_\phi$-dimensional space. HLS

**Definition 2.1** (HLS Representation). *A representation $\phi$ is* HLS *(the acronym refers to the last names of the authors of [16]) if*

$$\lambda^\star(\phi) := \lambda_{\min}\left(\mathbb{E}_{x \sim \rho}\left[\phi(x, a_x^\star)\phi(x, a_x^\star)^\mathsf{T}\right]\right) > 0$$

*where $\lambda_{\min}(A)$ denotes the minimum eigenvalue of a matrix A.*

Papini et al. showed that HLS, together with realizability, is a sufficient and necessary property for achieving constant regret in contextual stochastic linear bandits for non-redundant representations.

In order to deal with the general case where $\Phi$ may contain non-realizable representations, we rely on the following misspecification assumption from [11].

**Assumption 3** (Misspecification). *For each $\phi \notin \Phi^\star$, there exists $\epsilon_\phi > 0$ such that*

$$\min_{\theta \in \mathcal{B}_\phi} \min_{\pi:\mathcal{X}\to\mathcal{A}} \mathbb{E}_{x \sim \rho}\left[\left(\phi(x, \pi(x))^\mathsf{T}\theta - \mu(x, \pi(x))\right)^2\right] \geq \epsilon_\phi.$$

This assumption states that any non-realizable representation has a minimum level of misspecification on average over contexts and for any context-action policy. In the finite-context case, a sufficient condition for Asm. 3 is that, for each $\phi \notin \Phi^\star$, there exists a context $x \in \mathcal{X}$ with $\rho(x) > 0$ such that $\phi(x, a)^\mathsf{T}\theta \neq \mu(x, a)$ for all $a \in \mathcal{A}$ and $\theta \in \mathcal{B}_\phi$.

**Related work.** Several papers have focused on contextual bandits with an arbitrary function space to estimate the reward function under realizability assumptions [e.g., 4, 5, 7]. While these works consider a similar setting to ours, they do not aim to learn "good" representations, but rather focus on the exploration-exploitation problem to obtain sublinear regret guarantees. This often corresponds to recovering the maximum likelihood representation, which may not lead to the best regret. After the work in [11], the problem of representation learning with constant regret guarantees has also been studied in reinforcement learning [17, 18]. As these approaches build on the ideas in [11], they inherit the same limitations as [11].

Another related literature is the one of expert learning and model selection in bandits [e.g., 19–25], where the objective is to select the best candidate among a set of base learning algorithms or experts. While these algorithms are general and can be applied to different settings, including representation learning with a finite set of candidates, they may not be able to effectively leverage the specific structure of the problem. Furthermore, at the best of our knowledge, these algorithms suffers a polynomial dependence in the number of base algorithms ($|\Phi|$ in our setting) and are limited to worst-case regret guarantees. Whether the $\sqrt{T}$ or $\mathrm{poly}(|\Phi|)$ dependency can be improved in general is an open question (see [25] and [11, App. A]). Finally, [8, 26] studied the specific problem of model selection with nested linear representations, where the best representation is the one with the smallest dimension for which the reward is realizable.

Several works have recently focused on theoretical and practical investigation of contextual bandits with neural networks (NNs) [27–29]. While their focus was on leveraging the representation power of NNs to correctly predict the rewards, here we focus on learning representations with good spectral properties through a novel auxiliary loss. A related approach to our is [29] where the authors leverage self-supervised auxiliary losses for representation learning in image-based bandit problems.

## 3  A General Framework for Representation Learning

We introduce BANDITSRL (*Bandit Spectral Representation Learner*), an algorithm for stochastic contextual linear bandit that efficiently decouples representation learning from exploration-exploitation. As illustrated in Alg. 1, BANDITSRL has access to a fixed-representation contextual bandit algorithm $\mathfrak{A}$, the *base algorithm*, and it is built around two key mechanisms: ❶ a constrained optimization problem where the objective is to minimize a representation loss $\mathcal{L}$ to favor representations with HLS properties, whereas the constraint ensures realizability; ❷ a generalized likelihood ratio test (GLRT)

---

**Algorithm 1** BANDITSRL

---
1: **Input:** representations $\Phi$, no-regret algorithm $\mathfrak{A}$, confidence $\delta \in (0,1)$, update schedule $\gamma > 1$
2: Initialize $j = 0$, $\phi_j, \theta_{\phi_j,0}$ arbitrarily, $V_0(\phi_j) = \lambda I_{d_{\phi_j}}$, $t_j = 1$, let $\delta_j := \delta/(2(j+1)^2)$
3: **for** $t = 1, \ldots$ **do**
4:     Observe context $x_t$
5:     **if** $\mathrm{GLR}_{t-1}(x_t; \phi_j) > \beta_{t-1,\delta/|\Phi|}(\phi_j)$ **then**
6:         Play $a_t = \mathrm{argmax}_{a \in \mathcal{A}} \left\{ \phi_j(x_t, a)^\mathsf{T} \theta_{\phi_j, t-1} \right\}$ and observe reward $y_t$
7:     **else**
8:         Play $a_t = \mathfrak{A}_t(x_t; \phi_j, \delta_j/|\Phi|)$, observe reward $y_t$, and feed it into $\mathfrak{A}$
9:     **end if**
10:     **if** $t = \lceil \gamma t_j \rceil$ **and** $|\Phi| > 1$ **then**
11:         Set $j = j + 1$ and $t_j = t$
12:         Compute $\phi_j = \mathrm{argmin}_{\phi \in \Phi_t} \left\{ \mathcal{L}_t(\phi) \right\}$ and reset $\mathfrak{A}$
13:     **end if**
14: **end for**

---

to ensure that, if a HLS representation is learned, the base algorithm $\mathfrak{A}$ does not over-explore and the "good" representation is exploited to obtain constant regret.

**Mechanism ❶ (line 12).** The first challenge when provided with a generic set $\Phi$ is to ensure that the algorithm does not converge to selecting misspecified representations, which may lead to linear regret. This is achieved by introducing a hard constraint in the representation optimization, so that BANDITSRL only selects representations in the set (see also [11, App. F]),

$$\Phi_t := \left\{ \phi \in \Phi : \min_{\theta \in \mathcal{B}_\phi} E_t(\phi, \theta) \leq \min_{\phi' \in \Phi} \min_{\theta \in \mathcal{B}_{\phi'}} \left\{ E_t(\phi', \theta) + \alpha_{t,\delta}(\phi') \right\} \right\} \tag{1}$$

where $E_t(\phi, \theta) := \frac{1}{t} \sum_{s=1}^t \left( \phi(x_s, a_s)^T \theta - y_s \right)^2$ is the empirical mean-square error (MSE) of model $(\phi, \theta)$ and $\alpha_{t,\delta}(\phi) := \frac{40}{t} \log\left( \frac{8|\Phi|^2(12 L_\phi B_\phi t)^{d_\phi} t^3}{\delta} \right) + \frac{2}{t}$. This condition leverages the existence of a realizable representation in $\Phi_t$ to eliminate representations whose MSE is not compatible with the one of the realizable representation, once accounted for the statistical uncertainty (i.e., $\alpha_{t,\delta}(\phi)$).

Subject to the realizability constraint, the representation loss $\mathcal{L}_t(\phi)$ favours learning a HLS representation (if possible). As illustrated in Def. 2.1, a HLS representation is such that the expected design matrix associated to the optimal actions has a positive minimum eigenvalue. Unfortunately it is not possible to directly optimize for this condition, since we have access to neither the context distribution $\rho$ nor the optimal action in each context. Nonetheless, we can design a loss that works as a proxy for the HLS property whenever $\mathfrak{A}$ is a no-regret algorithm. Let $V_t(\phi) = \lambda I_{d_\phi} + \sum_{s=1}^t \phi(x_s, a_s)\phi(x_s, a_s)^\mathsf{T}$ be the empirical design matrix built on the context-actions pairs observed up to time $t$, then we define $\mathcal{L}_{\mathrm{eig},t}(\phi) := -\lambda_{\min}(V_t(\phi) - \lambda I_{d_\phi})/L_\phi^2$, where the normalization factor ensures invariance w.r.t. the feature norm. Intuitively, the empirical distribution of contexts $(x_t)_{t \geq 1}$ converges to $\rho$ and the frequency of optimal actions selected by a no-regret algorithm increases over time, thus ensuring that $V_t(\phi)/t$ tends to behave as the design matrix under optimal arms $\mathbb{E}_{x \sim \rho}[\phi(x, a_x^\star)\phi(x, a_x^\star)^\mathsf{T}]$. As discussed in Sect. 5 alternative losses can be used to favour learning HLS representations.

**Mechanism ❷ (line 5).** While Papini et al. [11] proved that LINUCB is able to exploit HLS representations, other algorithms such as $\epsilon$-greedy may keep forcing exploration and do not fully take advantage of HLS properties, thus failing to achieve constant regret. In order to prevent this, we introduce a *generalized likelihood ratio* test (GLRT). At each round $t$, let $\phi_{t-1}$ be the representation used at time $t$, then BANDITSRL decides whether to act according to the base algorithm $\mathfrak{A}$ with representation $\phi_{t-1}$ or fully exploit the learned representation and play greedily w.r.t. it. Denote by $\theta_{\phi,t-1} = V_{t-1}(\phi)^{-1} \sum_{s=1}^{t-1} \phi(x_s, a_s) y_s$ the regularized least-squares parameter at time $t$ for representation $\phi$ and by $\pi_{t-1}^\star(x; \phi) = \mathrm{argmax}_{a \in \mathcal{A}} \left\{ \phi(x, a)^\mathsf{T} \theta_{\phi, t-1} \right\}$ the associated greedy policy. Then, BANDITSRL selects the greedy action $\pi_{t-1}^\star(x_t; \phi_{t-1})$ when the GLR test is active, otherwise it selects the action proposed by the base algorithm $\mathfrak{A}$. Formally, for any $\phi \in \Phi$ and $x \in \mathcal{X}$, we define

the generalized likelihood ratio as

$$\text{GLR}_{t-1}(x;\phi) := \min_{a \neq \pi_{t-1}^\star(x;\phi)} \frac{\left(\phi(x,\pi_{t-1}^\star(x;\phi)) - \phi(x,a)\right)^\mathsf{T} \theta_{\phi,t-1}}{\|\phi(x,\pi_{t-1}^\star(x;\phi)) - \phi(s,a)\|_{V_{t-1}(\phi)^{-1}}} \tag{2}$$

and, given $\beta_{t-1,\delta}(\phi) = \sigma\sqrt{2\log(1/\delta) + d_\phi \log(1 + (t-1)L_\phi^2/(\lambda d_\phi))} + \sqrt{\lambda}B_\phi$, the GLR test is $\text{GLR}_{t-1}(x;\phi) > \beta_{t-1,\delta/|\Phi|}(\phi)$ [16, 30, 31]. If this happens at time $t$ and $\phi_{t-1}$ is realizable, then we have enough confidence to conclude that the greedy action is optimal, i.e., $\pi_{t-1}^\star(x_t;\phi_{t-1}) = a_{x_t}^\star$. An important aspect of this test is that it is run on the current context $x_t$ and it does not require evaluating global properties of the representation. While at any time $t$ it is possible that a non-HLS non-realizable representation may pass the test, the GLRT is sound as **1)** exploration through $\mathfrak{A}$ and the representation learning mechanism work in synergy to guarantee that *eventually* a realizable representation is always provided to the GLRT; **2)** only HLS representations are guaranteed to consistently trigger the test at any context $x$.

In practice, BANDITSRL does not update the representation at each step but in phases. This is necessary to avoid too frequent representation changes and control the regret, but also to make the algorithm more computationally efficient and practical. Indeed, updating the representation may be computationally expensive in practice (e.g., retraining a NN) and a phased scheme with $\gamma$ parameter reduces the number of representation learning steps to $J \approx \lceil \log_\gamma(T) \rceil$. The algorithm $\mathfrak{A}$ is reset at the beginning of a phase $j$ when the representation is selected and it is run on the samples collected during the current phase when the base algorithm is selected. If $\mathfrak{A}$ is able to leverage off-policy data, at the beginning of a phase $j$, we can warm-start it by providing $\phi_j$ and all the past data $(x_s, a_s, y_s)_{s \leq t_j}$. While the reset is necessary for dealing with *any* no-regret algorithm, it can be removed for algorithms such as LINUCB and $\epsilon$-greedy without affecting the theoretical guarantees.

**Comparison to LEADER.** We first recall the basic structure of LEADER. Denote by $\text{UCB}_t(x,a,\phi)$ the upper-confidence bound computed by LINUCB for the context-action pair $(x,a)$ and representation $\phi$ after $t$ steps. Then LEADER selects the action $a_t \in \arg\max_{a \in \mathcal{A}} \min_{\phi \in \Phi_t} \text{UCB}_t(x_t, a, \phi)$. Unlike the constrained optimization problem in BANDITSRL, this mechanism couples representation learning and exploration-exploitation and it requires optimizing a representation for the current $x_t$ and for each action $a$. Indeed, LEADER does not output a single representation and possibly chooses different representations for each context-action pair. While this enables LEADER to mix representations and achieve constant regret in some cases even when $\Phi$ does not include any HLS representation, it leads to two major drawbacks: **1)** the representation selection is directly entangled with the LINUCB exploration-exploitation strategy, **2)** it is impractical in problems where $\Phi$ is an infinite functional space (e.g., a deep neural network). The mechanisms ❶ and ❷ successfully address these limitations and enable BANDITSRL to be paired with any no-regret algorithm and to be scaled to any representation class as illustrated in the next section.

### 3.1 Extension to Neural Networks

We now consider a representation space $\Phi$ defined by the last layer of a NN. We denote by $\phi : \mathcal{X} \times \mathcal{A} \to \mathbb{R}^d$ the last layer and by $f(x,a) = \phi(x,a)^\mathsf{T}\theta$ the full NN, where $\theta$ are the last-layer weights. We show how BANDITSRL can be easily adapted to work with deep neural networks (NN).

*First*, the GLRT requires only to have access to the current context $x_t$ and representation $\phi_j$, i.e., the features defined by the last layer of the current network, and its cost is linear in the number of actions. *Second*, the phased scheme allows lazy updates, where we retrain the network only $\log_\gamma(T)$ times. *Third*, we can run any bandit algorithm with a representation provided by the NN, including LINUCB, LinTS, and $\epsilon$-greedy. *Fourth*, the representation learning step can be adapted to allow efficient optimization of a NN. We consider a regularized problem obtained through an approximation of the constrained problem:

$$\arg\min_\phi \left\{ \mathcal{L}_t(\phi) - c_{\text{reg}} \left( \min_{\phi',\theta'} \left\{ E_t(\phi',\theta') + \alpha_{t,\delta}(\phi') \right\} - \min_\theta E_t(\phi,\theta) \right) \right\}$$
$$= \arg\min_\phi \min_\theta \left\{ \mathcal{L}_t(\phi) + c_{\text{reg}} E_t(\phi,\theta) \right\}. \tag{3}$$

where $c_{\text{reg}} \geq 0$ is a tunable parameter. The fact we consider $c_{\text{reg}}$ constant allows us to ignore terms that do not depend on either $\phi$ or $\theta$. This leads to a convenient regularized loss that aims to minimize

the MSE (second term) while enforcing some spectral property on the last layer of the NN (first term). In practice, we can optimize this loss by stochastic gradient descent over a *replay buffer* containing the samples observed over time. The resulting algorithm, called NN-BANDITSRL, is a direct and elegant generalization of the theoretically-grounded algorithm.

While in theory we can optimize the regularized loss (3) with all the samples, in practice it is important to better control the sample distribution. As the algorithm progresses, we expect the replay buffer to contain an increasing number of samples obtained by optimal actions, which may lead the representation to solely fit optimal actions while increasing misspecification on suboptimal actions. This may compromise the behavior of the algorithm and ultimately lead to high regret. This is an instance of *catastrophic forgetting* induced by a biased/shifting sample distribution [e.g., 32]. To prevent this phenomenon, we store two replay buffers: *i)* an explorative buffer $\mathcal{D}_{\mathfrak{A},t}$ with samples obtained when $\mathfrak{A}$ was selected; *ii)* an exploitative buffer $\mathcal{D}_{\mathrm{glrt},t}$ with samples obtained when GLRT triggered and greedy actions were selected. The explorative buffer $\mathcal{D}_{\mathfrak{A},t}$ is used to compute the MSE $E_t(\phi, \theta)$. While this reduces the number of samples, it improves the robustness of the algorithm by promoting realizability. On the other hand, we use all the samples $\mathcal{D}_t = \mathcal{D}_{\mathfrak{A},t} \cup \mathcal{D}_{\mathrm{glrt},t}$ for the representation loss $\mathcal{L}(\phi)$. This is coherent with the intuition that mechanism ❶ works when the design matrix $V_t$ drifts towards the design matrix of optimal actions, which is at the core of the HLS property. Refer to App. C for a more detailed description of NN-BANDITSRL.

## 4 Theoretical Guarantees

In this section, we provide a complete characterization of the theoretical guarantees of BANDITSRL when $\Phi$ is a finite set of representations, i.e., $|\Phi| < \infty$. We consider the update scheme with $\gamma = 2$.

### 4.1 Constant Regret Bound for HLS Representations

We first study the case where a realizable HLS representation is available. For the characterization of the behavior of the algorithm, we need to introduce the following times:

- $\tau_{\mathrm{elim}}$: an upper-bound to the time at which all non-realizable representations are eliminated, i.e., for all $t \geq \tau_{\mathrm{elim}}$, $\Phi_t = \Phi^\star$;

- $\tau_{\mathrm{HLS}}$: an upper-bound to the time (if it exists) after which the HLS representation is selected, i.e., $\phi_t = \phi^\star$ for all $t \geq \tau_{\mathrm{HLS}}$, where $\phi^\star \in \Phi^\star$ is the unique HLS realizable representation;

- $\tau_{\mathrm{glrt}}$: an upper-bound to the time (if it exists) such that the GLR test triggers for the HLS representation $\phi^\star$ for all $t \geq \tau_{\mathrm{glrt}}$.

We begin by deriving a constant problem-dependent regret bound for BANDITSRL with HLS representations. The proof and explicit values of the constants are reported in App. B.[1]

**Theorem 4.1.** *Let $\mathfrak{A}$ be any no-regret algorithm for stochastic contextual linear bandits, $\Phi$ satisfy Asm. 1- 3, $|\Phi| < \infty$, $\gamma = 2$, and $\mathcal{L}_t(\phi) = \mathcal{L}_{\mathrm{eig},t}(\phi) := -\lambda_{\min}(V_t(\phi) - \lambda I_{d_\phi})/L_\phi^2$. Moreover, let $\Phi^\star$ contains a unique HLS representation $\phi^\star$. Then, for any $\delta \in (0,1)$ and $T \in \mathbb{N}$, the regret of BANDITSRL is bounded, with probability at least $1 - 4\delta$, as[2]*

$$R_T \leq 2\tau_{\mathrm{elim}} + \max_{\phi \in \Phi^\star} \overline{R}_{\mathfrak{A}}((\tau_{\mathrm{opt}} - \tau_{\mathrm{elim}}) \wedge T, \phi, \delta_{\log_2(\tau_{\mathrm{opt}} \wedge T)}/|\Phi|) \log_2(\tau_{\mathrm{opt}} \wedge T),$$

*where $\delta_j := \delta/(2(j+1)^2)$ and*

$$\tau_{\mathrm{opt}} = \tau_{\mathrm{glrt}} \vee \tau_{\mathrm{HLS}} \vee \tau_{\mathrm{elim}} \lesssim \tau_{\mathrm{alg}} + \frac{L_{\phi^\star}^2 \log(|\Phi|/\delta)}{\lambda^\star(\phi^\star)} \left( \frac{L_{\phi^\star}^2}{\lambda^\star(\phi^\star)} + \frac{d_{\phi^\star}}{\Delta^2} + \frac{d}{(\min_{\phi \notin \Phi^\star} \epsilon_\phi)\Delta} \right), \quad (4)$$

*with $\tau_{\mathrm{alg}}$ a finite (independent from the horizon $T$) constant depending on algorithm $\mathfrak{A}$ (see Tab. 1) and $\overline{R}_{\mathfrak{A}}(\tau, \phi, \delta)$ an anytime bound (non-decreasing in $\tau$ and $1/\delta$) on the regret accumulated over $\tau$ steps by $\mathfrak{A}$ using representation $\phi$ and confidence level $\delta$.*

---

[1]While Thm. 4.1 provides high-probability guarantees, we can easily derive a constant expected-regret bound by running BANDITSRL with a decreasing schedule for $\delta$ and with a slightly different proof.

[2]We denote by $a \wedge b$ (resp. $a \vee b$) the minimum (resp. the maximum) between $a$ and $b$.

The key finding of the previous result is that BANDITSRL achieves constant regret whenever a realizable HLS representation is available in the set $\Phi$, which may contain non-realizable as well as realizable non-HLS representations. The regret bound above also illustrates the "dynamics" of the algorithm and three main regimes. In the early stages, non-realizable representations may be included in $\Phi_t$, which may lead to suffering linear regret until time $\tau_{\text{elim}}$ when the constraint in the representation learning step filters out all non-realizable representations (first term in the regret bound). At this point, BANDITSRL leverages the loss $\mathcal{L}$ to favor HLS representations and the base algorithm $\mathfrak{A}$ to perform effective exploration-exploitation. This leads to the second term in the bound, which corresponds to an upper-bound to the sum of the regrets of $\mathfrak{A}$ in each phase in between $\tau_{\text{elim}}$ and $\tau_{\text{glrt}} \vee \tau_{\text{HLS}}$, which is roughly $\sum_{j_{\tau_{\text{elim}}} < j < j_{\tau_{\text{opt}}}} \overline{R}_{\mathfrak{A}}(t_{j+1} - t_j, \phi_j) \leq \max_{\phi \in \Phi^{\star}} \overline{R}_{\mathfrak{A}}(\tau_{\text{opt}} - \tau_{\text{elim}}, \phi) \log_2(\tau_{\text{opt}})$. In this second regime, in some phases the algorithm may still select non-HLS representations, which leads to a worst-case bound over all realizable representations in $\Phi^{\star}$. Finally, after $\tau_{\text{glrt}} \vee \tau_{\text{HLS}}$ the GLRT consistently triggers over time. During this last regime, BANDITSRL has reached enough accuracy and confidence so that the greedy policy of the HLS representation is indeed optimal and no additional regret is incurred.

We notice that the only dependency on the number of representations $|\Phi|$ in Thm. 4.1 is due to the rescaling of the confidence level $\delta \mapsto \delta/|\Phi|$. Since standard algorithms have a logarithmic dependence in $1/\delta$, this only leads to a logarithmic dependency in $|\Phi|$. On the other hand, due to the resets, BANDITSRL has an extra logarithmic factor in the effective regret horizon $\tau_{\text{opt}}$.

**Single HLS representation.** A noteworthy consequence of Thm. 4.1 is that any no-regret algorithm equipped with GLRT achieves constant regret when provided with a realizable HLS representation.

**Corollary 4.2.** *Let $\Phi = \Phi^{\star} = \{\phi^{\star}\}$ and $\phi^{\star}$ is HLS. Then, $\tau_{\text{elim}} = \tau_{\text{HLS}} = 0$ and, with probability at least $1 - 4\delta$, BANDITSRL suffers constant regret: $R_T \leq \overline{R}_{\mathfrak{A}}(\tau_{\text{glrt}} \wedge T, \phi^{\star}, \delta)$.*

This corollary also illustrates that the performance of $\mathfrak{A}$ is not affected when $\phi^{\star}$ is non-HLS (i.e., $\tau_{\text{glrt}} = \infty$), as BANDITSRL achieves the same regret of the base algorithm. Note that there is no additional logarithmic factor in this case since we do not need any reset for representation learning.

## 4.2 Additional Results

**No HLS representation.** A consequence of Thm. 4.1 is that when $|\Phi| > 1$ but no realizable HLS exists ($\tau_{\text{glrt}} = \infty$), BANDITSRL still enjoys a sublinear regret.

**Corollary 4.3** (Regret bound without HLS representation). *Consider the same setting in Thm. 4.1 and assume that $\Phi^{\star}$ does not contain any HLS representation. Then, for any $\delta \in (0, 1)$ and $T \in \mathbb{N}$, the regret of BANDITSRL is bounded, with probability at least $1 - 4\delta$, as follows:*

$$R_T \leq 2\tau_{\text{elim}} + \max_{\phi \in \Phi^{\star}} \overline{R}_{\mathfrak{A}}(T, \phi, \delta_{\log_2(T)}/|\Phi|) \log_2(T).$$

This shows that the regret of BANDITSRL is of the same order as the base no-regret algorithm $\mathfrak{A}$ when running with the worst realizable representation. While such worst-case dependency is undesirable, it is common to many representation learning algorithms, both in bandits and reinforcement learning [e.g. 4, 33].[3] In App. C, we show that an alternative representation loss could address this problem and lead to a bound scaling with the regret of the *best* realizable representation ($R_T \leq 2\tau_{\text{elim}} + \min_{\phi \in \Phi^{\star}} \overline{R}_{\mathfrak{A}}(T, \phi, \delta/|\Phi|) \log_2(T)$), while preserving the guarantees for the HLS case. Since the representation loss requires an upper-bound on the number of suboptimal actions and a carefully tuned schedule for guessing the gap $\Delta$, it is less practical than the smallest eigenvalue, which we use as the basis for our practical version of BANDITSRL.

**Algorithm-dependent instances and comparison to LEADER.** Table 1 reports the regret bound of BANDITSRL for different base algorithms. These results make explicit the dependence in the number of representations $|\Phi|$ and show that the cost of representation learning is only logarithmic. In the specific case of LINUCB for HLS representations, we highlight that the upper-bound to the time $\tau_{\text{opt}}$

---

[3]Notice that the worst-representation dependency is often hidden in the definition of $\Phi$, which is assumed to contain features with fixed dimension and bounded norm, i.e., $\Phi = \{\phi : \mathcal{X} \times \mathcal{A} \to \mathbb{R}^d, \sup_{x,a} \|\phi(x, a)\|_2 \leq L\}$. As $d$ and $B$ are often the only representation-dependent terms in the regret bound $\overline{R}_{\mathfrak{A}}$, no worst-representation dependency is reported.

| Algorithm | $\overline{R}_{\mathfrak{A}}(T, \phi, \delta/|\Phi|)$ | $\tau_{\mathrm{alg}}$ |
|---|---|---|
| LINUCB | $d_\phi^2 \log(|\Phi|T/\delta)^2/\Delta$ | $\frac{L_{\phi^\star}^2 d^2 \log(|\Phi|/\delta)^2}{\lambda^\star(\phi^\star)\Delta^2}$ |
| $\epsilon$-greedy with $\epsilon_t = t^{-1/3}$ | $\sqrt{d_\phi|\mathcal{A}|}\log(|\Phi|/\delta)T^{2/3}$ | $\frac{L_{\phi^\star}^6 (d|\mathcal{A}|)^{3/2} L^3 \log(|\Phi|/\delta)^3}{\lambda^\star(\phi^\star)^3 \Delta^3}$ |

Table 1: Specific regret bounds when using LINUCB or $\epsilon$-greedy as base algorithms. We omit numerical constants and logarithmic factors.

in Thm. 4.1 improves over the result of LEADER. While LEADER has no explicit concept of $\tau_{\mathrm{alg}}$, a term with the same dependence of $\tau_{\mathrm{alg}}$ in Tab. 1 appears also in the LEADER analysis. This term encodes an upper bound to the pulls of suboptimal actions and depends on the LINUCB strategy. As a result, the first three terms in Eq. 4 are equivalent to the ones of LEADER. The improvement comes from the last term ($\tau_{\mathrm{elim}}$), where, thanks to a refined analysis of the elimination condition, we are able to improve the dependence on the inverse minimum misspecification ($1/\min_{\phi \notin \Phi^\star} \epsilon_\phi$) from quadratic to linear (see App. B for a detailed comparison). On the other hand, BANDITSRL suffers from the worst regret among realizable representations, whereas LEADER scales with the *best* representation. As discussed above, this mismatch can be mitigated by using by a different choice of representation loss. In the case of $\epsilon$-greedy, the $T^{2/3}$ regret upper-bound induces a worse $\tau_{\mathrm{alg}}$ due to a larger number of suboptimal pulls. This in turns reflects into a higher regret to the constant regime. Finally, LEADER is still guaranteed to achieve constant regret by selecting different representations at different context-action pairs whenever non-HLS representations satisfy a certain mixing condition [cf. 11, Sec. 5.2]. This result is not possible with BANDITSRL, where one representation is selected in each phase. At the same time, it is the single-representation structure of BANDITSRL that allows us to accommodate different base algorithms and scale it to any representation space.

## 5 Experiments

We provide an empirical validation of BANDITSRL both in synthetic contextual linear bandit problems and in non-linear contextual problems [see e.g., 6, 27].

**Linear Benchmarks.** We first evaluate BANDITSRL on synthetic linear problems to empirically validate our theoretical findings. In particular, we test BANDITSRL with different base algorithms and representation learning losses and we compare it with LEADER.[4] We consider the "varying dimension" problem introduced in [11] which consists of six realizable representations with dimension from 2 to 6. Of the two representations of dimension $d = 6$, one is HLS. In addition seven misspecified representations are available. Details are provided in App. D. We consider LINUCB and $\epsilon$-greedy as base algorithms and we use the theoretical parameters, but we perform warm start using all the past data when a new representation is selected. Similarly, for BANDITSRL we use the theoretical parameters ($\gamma = 2$) and $\mathcal{L}_t(\phi) := \mathcal{L}_{\mathrm{eig},t}(\phi)$. Fig. 1 shows that, as expected, BANDITSRL with both base algorithms is able to achieve constant regret when a HLS representation exists. As expected from the theoretical analysis, $\epsilon$-greedy leads to a higher regret than LINUCB. Furthermore, empirically BANDITSRL with LINUCB obtains a performance that is comparable with the one of LEADER both with and without realizable HLS representation. Note that when no HLS exists, the regret of BANDITSRL with $\epsilon$-greedy is $T^{2/3}$, while LINUCB-based algorithms are able to achieve $\log(T)$ regret. When $\Phi$ contains misspecified representations (Fig. 1(center-left)), we can observe that in the first regime $[1, \tau_{\mathrm{elim}}]$ the algorithm suffers linear regret, after that we have the regime of the base algorithm ($[\tau_{\mathrm{elim}}, \tau_{\mathrm{glrt}} \vee \tau_{\mathrm{HLS}}]$) up to the point where the GLRT leads to select only optimal actions.

*Weak HLS.* Papini et al. [11] showed that when realizable representations are redundant (i.e., $\lambda^\star(\phi^\star) = 0$), it is still possible to achieve constant regret if the representation is "weakly"-HLS, i.e., the features of the optimal actions span the features $\phi(x, a)$ associated to any context-action pair, but not necessarily $\mathbb{R}^{d_\phi}$. To test this case, we pad a 5-dimensional vector of ones to all the features of the six realizable representations in the previous experiment. To deal with the weak-HLS condition, we introduce the alternative representation loss $\mathcal{L}_{\mathrm{weak},t}(\phi) = -\min_{s \leq t} \{\phi(x_s, a_s)^\mathsf{T}(V_t(\phi) - \lambda I_{d_\phi})\phi(x_s, a_s)/L_\phi^2\}$. Since, $V_t(\phi) - \lambda I_{d_\phi}$ tends to behave as $\mathbb{E}_x[\phi^\star(x)\phi^\star(x)^\mathsf{T}]$, this loss encourages representations where all the observed features are spanned by the optimal arms, thus promoting weak-HLS representations

---

[4]We do not report the performance of model selection algorithms. An extensive analysis can be found in [11], where the author showed that LEADER was outperforming all the baselines.

(see App. C for more details). As expected, Fig. 1(right) shows that the min-eigenvalue loss $\mathcal{L}_{\text{eig},t}$ fails in identifying the correct representation in this domain. On the other hand, BANDITSRL with the novel loss is able to achieve constant regret and converge to constant regret (we cut the figure for readability), and behaves as LEADER when using LINUCB.

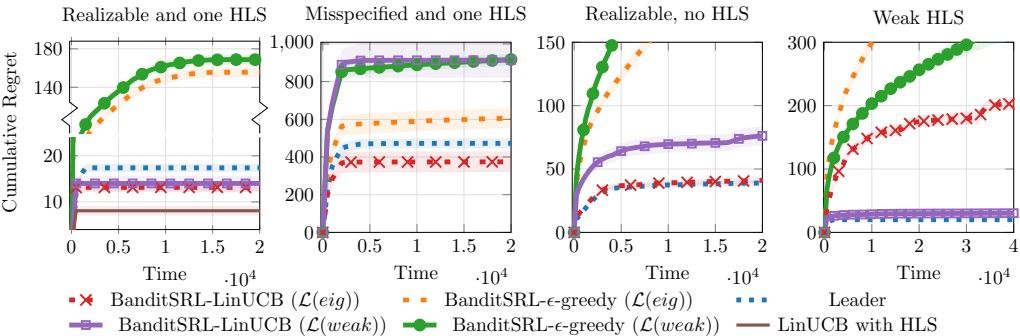

Figure 1: Varying dimension experiment with all realizable representations (left), misspecified representations (center-left), realizable non-HLS representations (center-right) and weak-HLS (right). Experiments are averaged over 40 repetitions.

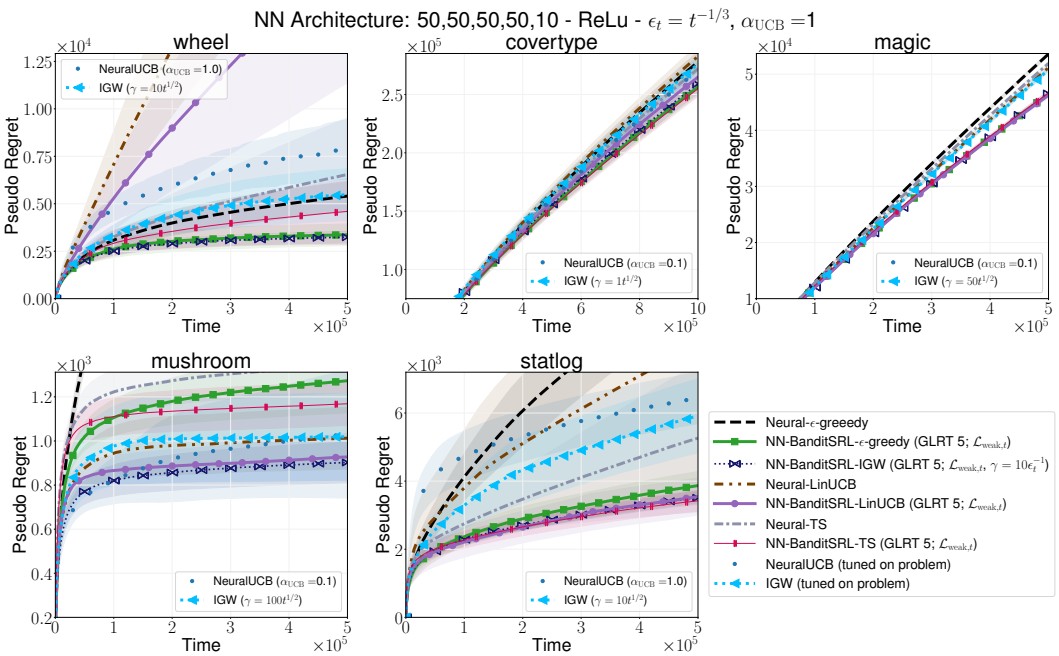

Figure 2: Average cumulative regret (over 20 runs) in non-linear domains.

**Non-Linear Benchmarks.** We study the performance of NN-BANDITSRL in classical benchmarks where non-linear representations are required. The code is available at the following URL. We only consider the weak-HLS loss $\mathcal{L}_{\text{weak},t}(\phi)$ as it is more general than full HLS. As base algorithms we consider $\epsilon$-greedy and inverse gap weighting (IGW) with $\epsilon_t = t^{-1/3}$, and LINUCB and LINTS with theoretical parameters. These algorithms are run on the representation $\phi_j$ provided by the NN at each phase $j$. We compare NN-BANDITSRL against the base algorithms using the maximum-likelihood representation (i.e., Neural-($\epsilon$-greedy, LINTS) [6] and Neural-LINUCB [28]), supervised learning with the IGW strategy [e.g., 7, 10] and NeuralUCB [27][5] See App. C-D for details.

---

[5]For ease of comparison, all the algorithms use the same phased schema for fitting the reward and recomputing the parameters. NeuralUCB uses a diagonal approximation of the design matrix.

In all the problems[6] the reward function is highly non-linear w.r.t. contexts and actions and we use a network composed by layers of dimension $[50, 50, 50, 50, 10]$ and ReLu activation to learn the representation (i.e., $d = 10$). Fig. 2 shows that all the base algorithms ($\epsilon$-GREEDY, IGW, LIN-UCB, LINTS) achieve better performance through representation learning, outperforming the base algorithms. This provides evidence that NN-BANDITSRL is effective even beyond the theoretical scenario.

For the baseline algorithms (NEURALUCB, IGW) we report the regret of the best configuration on each individual dataset, while for NN-BANDITSRL we fix the parameters across datasets (i.e., $\alpha_{\mathrm{GLRT}} = 5$). While this comparison clearly favours the baselines, it also shows that NN-BANDITSRL is a robust algorithm that behaves better or on par with the state-of-the-art algorithms. In particular, NN-BANDITSRL uses theoretical parameters while the baselines use tuned configurations. Optimizing the parameters of NN-BANDITSRL is outside the scope of these experiments.

## 6  Conclusion

We proposed a novel algorithm, BANDITSRL, for representation selection in stochastic contextual linear bandits. BANDITSRL combines a mechanism for representation learning that aims to recover representations with good spectral properties, with a generalized likelihood ratio test to exploit the recovered representation. We proved that, thanks to these mechanisms, BANDITSRL is not only able to achieve sublinear regret with any no-regret algorithm $\mathfrak{A}$ but, when a HLS representation exists, it is able to achieve constant regret. We demonstrated that BANDITSRL can be implemented using NNs and showed its effectiveness in standard benchmarks.

A direction for future investigation is to extend the approach to a weaker misspecification assumption than Asm. 3. Another direction is to leverage the technical and algorithmic tools introduced in this paper for representation learning in reinforcement learning, e.g., in low-rank problems [e.g. 38].

## Acknowledgments and Disclosure of Funding

M. Papini was supported by the European Research Council (ERC) under the European Union's Horizon 2020 research and innovation programme (Grant agreement No. 950180).

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
