# OpenReview forum: "Scalable Representation Learning in Linear Contextual Bandits with Constant Regret Guarantees"
_NeurIPS.cc/2022/Conference — NeurIPS 2022 Accept_

### Official Review · Reviewer_8KSa · 2022-06-19

**Rating:** 7
**Confidence:** 3
**Soundness:** 4 excellent
**Presentation:** 4 excellent
**Contribution:** 3 good

**Summary:**

This paper provides better algorithms to learn the good representation from a set of realizable representation in contextual bandits. The algorithm is interesting and enjoys provable guarantees. Experiments are also conducted to justify the performance.

**Questions:**

1. HLS condition is never cited. I think that is from the last name of the authors of "Adaptive exploration in linear contextual bandit" and should be acknowledged.

**Ethics Review Area:**

["I don’t know"]

**Limitations:**

I feel in experiments the authors could try to discuss some failure cases and when people should be cautious to use this algorithm.

**Strengths And Weaknesses:**

This paper is well-written and well-organized. The algorithm is interesting and can be generalized with neural net function approximation. I appreciate the author could carefully compare with prior LEADER algorithms.

I only have some questions for the experiments part, especially  for the Non-Linear Benchmarks. For those datasets statlog, magic, covertype, it looks like the best proposed algorithm can achieve constant regret easily. This is a huge improvement over the class of TS reported in "DEEP BAYESIAN BANDITS SHOWDOWN". Can you comment why this could happen and do you make a fair comparsion? I can understand approximate TS does not do representation learning but is your result suggesting HLS is satisfied in those dataset? Do you cherry-pick those dataset? I feel this is important to understand how frequently HLS conditions can be satisfied in real world problems.

---

> ### Author Response · Authors · 2022-08-01
> **Answer to Reviewer 8KSa**
>
> Thank you for the review and the positive feedback.
>
> **Point 1 (dataset-based experiments):** While cleaning the code after the deadline, we realized that the experiments based on datasets (i.e. statlog, magic, covertype) reported in the old paper were not run on the whole dataset but only on a small subset of contexts (the other experiments are unaffected, i.e., wheel and the theoretical ones). This led to considering a much simpler problem, where the network has enough capacity to trivially achieve realizability. We then rerun the code using all the contexts  (and added the mushroom dataset) and the new results lead to the same conclusions: NN-BanditSRL is effective in practice and outperforms state-of-the-art baselines. Notice that in this case NN-BanditSRL is clearly not operating in the realizable regime and the problems are misspecified (especially in the case of covertype and magic). We think the two sets of experiments (old and new) are complementary and offer a more complete analysis of the behavior of the algorithms. Please refer to the updated version of the paper where we have added the new experiments, the old experiments are in the appendix. Changes are highlighted in blue.
>
> Figure 2 in the updated paper shows that NN-BanditSRL behaves better than the baselines algorithms (that now include NeuralUCB [1], inverse gap weighting exploration [2,3], random Fourier features and Neural-LinTS [4]). Since no guidance is provided on how to set the parameters of the baselines, we do hyperparameter tuning for each of them on each individual dataset. Thus the results we obtain should be intended as an upper-bound to the performance of the baselines. On the other hand, for NN-BanditSRL we fix the parameters across datasets (i.e., $\alpha_{GLRT} =5$). In the appendix we show that the same conclusion holds across almost all the configurations considered. We will release the code for reproducibility.
>
> **Point 2 (HLS acronym):** Thank you for pointing this out. We referred to [5] where, as far as we know, this name was first used. In the revised paper, we explicitly mention that the term refers to the initials of the authors of “Adaptive exploration in linear contextual bandit”.
>
> **Point 3 (failure mode):** We agree with the reviewer that it is useful to illustrate failure modes. In general,  we noticed that our algorithm is robust to change in the parameters (see ablation in the appendix). The only case that is critical for NN-BanditSRL is when the GLRT parameter $\alpha_{GLRT}$ is too small and there is misspecification. Indeed, in the new experiments where misspecification is relevant and there is no realizable model, if the scaling parameter is too small, the GLRT may trigger too often, leading to under-exploration and worse regret (see Appendix D.2.1). Indeed, in the agnostic case the GLRT is not correct and may trigger erroneously, forcing the algorithm to act greedily and preventing exploration. Furthermore, since we do not use GLRT samples in the MSE loss, these samples will not contribute to improving the precision of the model, causing further misspecification. As a result, in practice we recommend using a relatively large value for $\alpha_{GLRT}$ (e.g., 5) since it will make the test more robust. While this may cause some extra unnecessary exploration when NN-BanditSRL is paired with algorithms such as eps-greedy, it is a conservative choice that worked well across all problems and datasets in our experiments. On the other hand, when paired with exploration-efficient algorithms such as LinUCB a large value of $\alpha_{GLRT}$ has a much milder impact on the overall performance. Another important dimension when designing the algorithm is to favor deeper vs wider networks. As we discussed in the paper, while wider networks may allow for a good fitting of the value function they tend to leave very limited space for optimizing the representation towards the HLS condition. On the other hand, a deeper network has more capacity in transforming the input into an alternative representation that we can more effectively skew towards HLS through the regularization term.
>
>
> [1] Dongruo Zhou, Lihong Li, Quanquan Gu: Neural Contextual Bandits with UCB-based Exploration. ICML 2020: 11492-11502
>
> [2] Dylan J. Foster, Alexander Rakhlin: Beyond UCB: Optimal and Efficient Contextual Bandits with Regression Oracles. ICML 2020: 3199-3210
>
> [3] David Simchi-Levi, Yunzong Xu: Bypassing the Monster: A Faster and Simpler Optimal Algorithm for Contextual Bandits under Realizability. CoRR abs/2003.12699 (2020)
>
> [4] Carlos Riquelme, George Tucker, Jasper Snoek: Deep Bayesian Bandits Showdown: An Empirical Comparison of Bayesian Deep Networks for Thompson Sampling. ICLR (Poster) 2018
>
> [5] Matteo Papini, Andrea Tirinzoni, Marcello Restelli, Alessandro Lazaric, Matteo Pirotta: Leveraging Good Representations in Linear Contextual Bandits. ICML 2021: 8371-8380

---

### Official Review · Reviewer_Jibx · 2022-07-07

**Rating:** 7
**Confidence:** 3
**Soundness:** 4 excellent
**Presentation:** 4 excellent
**Contribution:** 3 good

**Summary:**

This paper proposes a new algorithm BanditSRL, that can select the realizable HLS representations from a family of representations for linear contextual bandit learning. The new algorithm is a framework that can apply to any constant-regret base algorithms like LinearUCB. Theoretical analysis show that the algorithm can indeed achieve constant regret when an HLS representation is present and sub-linear regret when only realizable representations are available. The algorithm is also extended to a more practical version that utilizes deep neural networks as the representation family. Experiments show the algorithms are efficient and can achieve small regret.

**Questions:**

At Line 204, Equation (2), what exactly does the Lagrangian relaxation refer to? Here the equality sign is drawn but I cannot easily see why the equality can hold. Could you explain in detail how this is derived? Or is this an approximation.

**Limitations:**

There are no foreseeable negative societal impact.

I think the authors should also compare the regret bounds in detail with LEADER in terms of all the variables. While the proposed algorithm BanditSRL surely is more general than LEADER, potentially the regret bound dependence can be worse due to a more complicated analysis.

**Strengths And Weaknesses:**

Strength:
+ This paper proposes a new solution to representation learning for linear contextual bandit. Compared with previous work, the new algorithm has a very clean structure and can utilize any base linear bandit algorithm. I believe the algorithmic design and the theoretical analysis are quite original.
+ The paper is written in high quality. It is clearly written and easy to follow. Potential concerns are discussed and addressed.
+ The improvement compared with Papini et al. can be significant in that the algorithm goes beyond just LinUCB and the NN-based variant can be practical.

---

> ### Author Response · Authors · 2022-08-01
> **Answer to Reviewer Jibx**
>
> Thank you for the review and the positive feedback.
>
> **Point 1 (regularized objective):** The regularized formulation considered in NN-BanditSRL has to be intended as an approximation. The objective is to  derive a practical approach directly from the theoretical algorithm with an approximation. In practice, the Lagrangian parameter $c_{reg}$ is a hyperparameter. In our experiments a default value of 1 worked well across all problems and datasets. The equality in Eq. (3) comes from ignoring the terms that do not depend on either $\phi$ or $\theta$ (given that $c_{reg}$ is fixed), since they do not affect the value of the $\arg\min$. We have clarified this in the revised version of the paper (changes are highlighted in blue).
>
>
> **Point 2 (comparison with LEADER):**  Let us compare BanditSRL using LinUCB as the base algorithm with LEADER in the same setting as in Theorem 6.1 of the current paper: among the realizable representations in $\Phi$, there exists a unique one (call it $\phi^\star$) which is HLS.
>
> The regret bound of BanditSRL can be extracted from the general bound in Theorem 6.1 by instantiating the quantities $\tau_{\mathrm{alg}}$ (and $\overline{R}_{\mathfrak{A}}$) for LinUCB as in Table 1.
>
> Letting $d_{\max}$ and $L_{\max}$ denote the maximum feature dimensionality and norm across all representations in $\Phi$, we get
>
> $R_T \lesssim \frac{d_{\max}\log(|\Phi|/\delta)}{\min_{\phi\notin \Phi^\star}\epsilon_\phi} + \frac{d_{\max}^2 \log(|\Phi|\tau / \delta)^2}{\Delta}$
>
> where
>
> $\tau \lesssim \frac{L_{\phi^\star}^2 d_{\max}^2\log(|\Phi|/\delta)^2}{\lambda^\star(\phi^\star)\Delta^2} + \frac{L_{\phi^\star}^2\log(|\Phi|/\delta)}{\lambda^\star(\phi^\star)} \left( \frac{L_{\phi^\star}^2}{\lambda^\star(\phi^\star)} + \frac{d_{\phi^\star}}{\Delta^2} + \frac{d_{\max}}{(\min_{\phi\notin\Phi^\star}\epsilon_\phi)\Delta} \right)$
>
> Similarly, denoting $d_{\min}$ the minimum feature dimensionality in $\Phi$, we can extract the following regret bound for LEADER from Theorem 2 and Appendix F of Papini et al. (2021)
>
> $R_T \lesssim \frac{d_{\max}\log(|\Phi|/\delta)}{\min_{\phi\notin \Phi^\star}\epsilon_\phi^2} + \frac{d_{\min}^2 \log(|\Phi|\tau / \delta)^2}{\Delta}$
>
> where
>
> $\tau \lesssim \frac{L_{\phi^\star}^2 d_{\min}^2\log(|\Phi|/\delta)^2}{\lambda^\star(\phi^\star)\Delta^2} + \frac{L_{\phi^\star}^2\log(|\Phi|/\delta)}{\lambda^\star(\phi^\star)} \left( \frac{L_{\phi^\star}^2}{\lambda^\star(\phi^\star)} + \frac{d_{\phi^\star}}{\Delta^2} + \frac{d_{\max}}{(\min_{\phi\notin\Phi^\star}\epsilon_\phi^2)\Delta} \right)$
>
> We note that the regret bound for LEADER scales, in some terms, with the minimum feature dimensionality $d_{\min}$, while BanditSRL scales with the maximum one $d_{\max}$. As the reviewer suggests, this is the price we have to pay for generality and scalability. In fact, LEADER is able to achieve such a dependence by playing at each step the action with minimum (across representations) upper confidence bound. This, in turn, requires fitting a different representation for each context-arm pair, which clearly does not scale in practice.
>
> We further note that the regret bound of BanditSRL features an improved dependence on the minimum misspecification w.r.t. the one of LEADER (from $\epsilon_\phi^2$ to $\epsilon_\phi$). This is thanks to our refined concentration bounds for the mean-square error.

---

> > ### Comment · Reviewer_Jibx · 2022-08-07
> > **Thanks for the response**
> >
> > I have read the response and my concerns are addressed. My rating will remain unchanged.

---

### Official Review · Reviewer_MyZk · 2022-07-11

**Rating:** 7
**Confidence:** 4
**Soundness:** 3 good
**Presentation:** 4 excellent
**Contribution:** 3 good

**Summary:**

This paper studies representation learning for stochastic contextual linear bandit problems. This is done by combining a novel constrained optimization problem to learn a realizable representation with a generalized likelihood ratio test to exploit the structural information. The authors propose BANDITSRL algorithm which could be paired with any no-regret algorithm and achieve constant regret.


**Questions:**

See above.

**Limitations:**

Yes.

**Strengths And Weaknesses:**

Pros: This paper focuses on a very important question and it explores the structural information contained in the learning problem. It can be viewed as a combination of model-based and model-free learning, which is practically promising and theoretically plausible.

Cons: It is not clear how exploration plays a role here and how it could be improved by the  BANDITSRL algorithm. The benefit of finding realizable representations is that representations with certain spectral properties may be more effective for the exploration-exploitation trade-off. In particular, [1] showed that when the regularity condition (Assumption 2 in this paper) is satisfied, the greedy algorithm (with no exploration) is optimal for the linear contextual bandit problem since the feature itself has certain exploratory properties. I am wondering why this does not hold for the feature-based linear contextual bandit.

In addition, this representation formulation seems to be closely related to the sparse linear contextual bandits problem by replacing the linear contexts with feature contexts. The authors are encouraged to explore the connections and differences.

[1] Ren, Zhimei, and Zhengyuan Zhou. "Dynamic batch learning in high-dimensional sparse linear contextual bandits." arXiv preprint arXiv:2008.11918 (2020).


==========
I have read the reviews from other reviewers and the responses from the authors. The authors have addressed my concerns and I am happy to keep my recommendation to "accept" this submission!

---

> ### Author Response · Authors · 2022-08-01
> **Answer to Reviewer MyZk**
>
> Thank you for the review and the positive feedback.
>
> **Point 1 (connection with Assumption2 in [1]):** Papini et al. 2021 showed that HLS is necessary (and sufficient) for any algorithm to achieve *constant regret* in the feature-based setting. However, HLS may not be sufficient to achieve logarithmic regret *without exploration* (i.e., with a greedy algorithm) and a stronger condition (as shown in [1,2]) may be necessary in this case.
>
> Concerning Assumption 2 in [1], note that it is related to Assumption 4 (Positive-Definiteness) in [2] (which is equivalent to Assumption F.2 in [1]). Both conditions were used to prove logarithmic regret for the greedy algorithm. The only difference is due to the joint or disjoint setting (called C-model and P-model in [1]). Papini et al. [3] provided a characterization of the numerous diversity conditions proposed in the literature, including conditions like the one mentioned by the reviewer. In particular, they showed that conditions like Assumption 4 (i.e. Assumption 2 in [1] in the C-model) are stronger than the HLS condition (see Lemma 1).
>
> Formally, let’s extend Assumption 2 in [1] to feature-based stochastic linear contextual bandits:
>
> $\forall v, \theta ~: P\big(v^\mathsf{T} \phi^\star_{\theta}(x){\phi^\star_{\theta}(x)}^\mathsf{T} v > \gamma\big) > \rho > 0$
>
> Here $\phi^\star_{\theta}(x) := \phi(x,\arg\max_{a}\phi(x,a)^T \theta)$ denotes the optimal feature at context $x$ induced by parameter $\theta$.
>
> With this assumption, the same results derived in [1] would hold for the feature-based problem. This assumption is stronger than HLS since it requires a HLS-like condition to hold for any greedy policy ($\forall \theta$, i.e. any possible linear reward in the features). Formally, to connect Assumption 2 in [1] to the HLS condition, we can see that by Markov inequality and algebraic manipulations:
>
> $\forall v, \theta,~~P\big(v^\mathsf{T} \phi^\star_{\theta}(x) {\phi^\star_{\theta}(x)}^\mathsf{T} v > \gamma\big) > \rho >0$
>
> $\implies \forall v, \theta,~~ E_{x} [v^T \phi^\star_{\theta}(x) {\phi^\star_{\theta}(x)}^\mathsf{T} v] > \rho \gamma >0$ (Markov inequality)
>
> $\implies\forall \theta,~~ \min_{v, \|v\|=1} v^T E_{x} [\phi^\star_{\theta}(x) {\phi^\star_{\theta}(x)}^\mathsf{T}] v > \rho\gamma >0$
>
> $\implies\forall \theta,~~ \lambda_{\min} E_{x}[\phi^\star_{\theta}(x) {\phi^\star_{\theta}(x)}^\mathsf{T}] > \rho\gamma >0$
>
> The HLS assumption is much milder since the condition above is required only for the optimal policy of the considered problem, i.e., only for the parameter $\theta^\star$.
>
> **Point 2 (sparse bandits):** The connection between sparse representations and HLS is an interesting question, but note that enforcing HLS does not necessarily encourage sparse or low-dimensional representations. As shown in the varying-dimension experiment by Papini et al. (2021), which we revisited in our paper, a higher-dimensional HLS representation can lead to significantly smaller regret than a lower-dimensional non-HLS representation. Still, the interaction between algorithms designed to exploit sparsity and HLS representations is an interesting topic for future work.
>
> [1] Zhimei Ren, Zhengyuan Zhou: Dynamic Batch Learning in High-Dimensional Sparse Linear Contextual Bandits. CoRR abs/2008.11918 (2020)
>
> [2] Hamsa Bastani, Mohsen Bayati, Khashayar Khosravi: Mostly Exploration-Free Algorithms for Contextual Bandits. Manag. Sci. 67(3): 1329-1349 (2021)
>
> [3] Matteo Papini, Andrea Tirinzoni, Marcello Restelli, Alessandro Lazaric, Matteo Pirotta: Leveraging Good Representations in Linear Contextual Bandits. ICML 2021: 8371-8380

---

### Meta-Review · Area_Chair_XctH · 2022-08-27

**Recommendation:** Accept
**Confidence:** Certain

**Metareview:**

All reviewers are in agreement that this paper provides better algorithms to learn good representation from a set of realizable representation in contextual linear bandits. The proposed algorithm is deemed to be novel and general. Accept.

In the final version, please release the code for reproducibility. The authors are encouraged to include the detailed comparison with LEADER in the rebuttal.

**Award:**

No

---

### Decision · Program_Chairs · 2022-09-14

Accept